# Extracellular Vesicles Profiling in Acute Myeloid Leukemia Cell Lines: A Proteomic Characterization

**DOI:** 10.3390/cells14211651

**Published:** 2025-10-22

**Authors:** Beatrice Dufrusine, Maria Concetta Cufaro, Alice Di Sebastiano, Erika Pizzinato, Pina Nardinocchi, Ilaria Cicalini, Serena Pilato, Antonella Fontana, Damiana Pieragostino, Enrico Dainese, Luca Federici

**Affiliations:** 1Department of Bioscience and Technology for Food Agriculture and Environment, University of Teramo, 64100 Teramo, Italy; bdufrusine@unite.it (B.D.); pnardinocchi@unite.it (P.N.); edainese@unite.it (E.D.); 2Center for Advanced Studies and Technology (CAST), University “G. d’Annunzio” of Chieti-Pescara, 66100 Chieti, Italy; maria.cufaro@unich.it (M.C.C.); alice.disebastiano@phd.unich.it (A.D.S.); erika.pizzinato@unidav.it (E.P.); ilaria.cicalini@unich.it (I.C.); damiana.pieragostino@unich.it (D.P.); 3Department of Innovative Technologies in Medicine and Dentistry, University “G. d’Annunzio” of Chieti-Pescara, 66100 Chieti, Italy; 4Department of Human, Legal and Economic Sciences, Telematic University ‘Leonardo da Vinci’, 66100 Chieti, Italy; 5Department of Pharmacy, University “G. dAnnunzio” of Chieti-Pescara, 66100 Chieti, Italy; serena.pilato@unich.it (S.P.); afontana@unich.it (A.F.)

**Keywords:** acute myeloid leukemia, extracellular vesicles, EV-mediated signaling, proteomics, cathepsins

## Abstract

Extracellular vesicles (EVs) express features of parental cells and are fundamental in modulating the crosstalk between cancer cells and their environment. Increasing evidence suggests that EVs have a pivotal role in tumorigenesis, cancer development, and drug resistance. EVs are also involved in controlling the communication between hematopoietic stem cells and the surrounding microenvironment in the bone marrow (BM), during several processes such as self-renewal, mobilization, and lineage differentiation. Proteins expressed in cancer cell-derived EVs can be useful to further understand the regulation of hematopoietic stem cell fate, a fundamental mechanism in acute myeloid leukemia (AML). Furthermore, EVs are implicated in transmitting drug-resistance mechanisms in solid and not-solid cancer types. Here, using a proteomic approach, we analyze and validate the protein profile of EVs from three AML cell lines with different genotypes, namely OCI-AML-2, OCI-AML-3, and HL-60. The majority of the identified proteins were significantly enriched in the Gene Ontology category ‘Extracellular Exosome’. Network model analysis of EV proteins revealed several significantly modulated pathways, including inflammation activation and metastatic processes in AML cell-derived EVs. The EVs proteomic profiling allows us to identify the EVs-associated molecules and pathways that could impact cancer progression and drug resistance.

## 1. Introduction

Acute myeloid leukemia (AML) is a cancer of blood and bone marrow (BM); it is the most common form of acute leukemia in adults and is characterized by uncontrolled proliferation of myeloid progenitor cells [1,2]. The 5-year relative survival rate changes with age, and for people over age 60 is 15% [3]. In hematological cancers, tumor cell-derived extracellular vesicles (EVs) concur to support cancer cell growth by modulating several processes such as angiogenesis, metastatization, metabolic reprogramming, and functional changes in microenvironment resident cells [4,5]. EVs are an atherogenic type of cell-derived vesicles including exosomes (40–120 nm) and macrovesicles (100–1000 nm) [4,6]. EVs are lipid bilayer-delimited particles enriched in tetraspanin proteins on their surface such as CD9, CD63, and CD81, which are commonly used as EVs biomarkers [7]. EVs mediate cellular communication not reliant on direct cell-to-cell contact, by transporting several types of molecules such as lipids, proteins, and nucleic acids [6,8]. In particular, EVs have been reported to be fundamental in the crosstalk between leukemia cells and microenvironment cells in the BM niche [5,9]. Leukemia cells control molecular changes in stromal, endothelial cells as well as in mesenchymal progenitors in the BM niche to generate a growth-permissive microenvironment [5,10]. Furthermore, AML patients’ sera have shown higher levels of EVs, and their content might be predictive as to the response to chemotherapy and drug resistance [11,12,13]. To further characterize the role of EVs in AML and to identify possible molecular targets useful to disease treatment, here we used a proteomic approach. We selected three cell lines commonly used as a model for AML, namely OCI-AML-2, OCI-AML-3, and HL-60, which display different genotypes. OCI-AML-2 is characterized by methyltransferase 3 A (DNMT3A) mutation occurring in about 20–30% of cases with AML, responsible for an abnormal DNA methylation pattern and associated with poor outcomes in AML [14,15]. OCI-AML-3 cells harbor mutation of the nucleolar protein nucleophosmin 1 (NPM1), reported in approximately 30–35% AML cases, leading to its aberrant cytoplasmatic localization [16,17,18]. Instead, HL-60 cells are null for p53 expression and carry amplification of the c-MYC proto-oncogene, leading to malignant transformation [19,20]. These different genotypic characteristics make each of these cell lines a specific model for investigating different aspects of AML disease. We proceeded with EVs isolation using the ultracentrifugation method and we performed EVs proteomic analysis. Our collective data suggest that AML cell-derived EVs could be involved in the modulation of molecular pathways essential in leukemia disease.

## 2. Materials and Methods

### 2.1. AML Cells Culture and Cell-Derived Extracellular Vesicles Purification

OCI-AML-2 and OCI-AML-3 cells were routinely grown in Eagle’s Minimum Essential Medium (Gibco, Life Technologies, Toulouse, France) supplemented with 20% FBS (Gibco, Life Technologies, Toulouse, France), penicillin (100 U/mL) and streptomycin (100 mg/mL) (Gibco, Life Technologies, Toulouse, France) at 37 °C in 5% CO_2_. HL-60 cells were grown in Iscove’s Modified Dulbecco’s Medium (Gibco, Life Technologies, Toulouse, France), supplemented with 20% FBS (Gibco, Life Technologies, Toulouse, France), penicillin (100 U/mL), and streptomycin (100 mg/mL) (Gibco, Life Technologies, Toulouse, France) at 37 °C in 5% CO_2_. As concerns cell-derived EVs isolation by ultracentrifugation, AML cells were cultivated for 48 h in serum-free medium. Around 100 mL of cell suspension was collected and differential ultracentrifugation was performed for EVs isolation [21]. Briefly, supernatant was centrifuged at 300, 2000, 10,000, and 100,000× *g* for 10, 30, and 70 min, respectively, at 4 °C. The pellet of the last centrifugation consisted of crude extracellular vesicles, which was analyzed by Western Blotting for protein targets and classical exosomal markers expression.

### 2.2. Dynamic Light Scattering Analysis

The size and ζ-potential values of the EVs derived from OCI-AML-2, OCI-AML-3, and HL-60 cells were measured using a 90Plus/BI-MAS ZetaPlus to analyze multiangle particle size (Brookhaven Instruments Corp., Holtsville, NY, USA). For size measurements, the autocorrelation function of the scattered light was analyzed, assuming a log Gaussian distribution of the vesicle size. The mean size and polydispersity index (PDI) were obtained. The ζ-potential values were calculated from the electrophoretic mobility by means of the Helmholtz–Smoluchowski relationship.

### 2.3. Western Blotting Analysis

Cells and purified EVs lysis were performed as follows: whole cell lysate was lysed in RIPA buffer containing protease and phosphatase inhibitors (Sigma Aldrich Corporation, St. Louis, MO, USA), then clarified by centrifugation at 13,000 rpm for 10 min at 4 °C; meanwhile, isolated EVs were prepared in a reducing (for protein targets and actin immunoblotting) or non-reducing (for CD9 and CD63 immunoblotting, without 2-mercaptoethanol) sample buffer (50 mM Tris-HCl pH 6.8, 5% glycerol, 2% SDS, 1.5% 2-mercaptoethanol with bromophenol blue) and heated at 95 °C for 10 min. Cellular protein amount of cellular and EVs lysates was determined using the Bradford reagent (Bio-Rad, Berkeley, CA, USA), and 15 µg of proteins were separated on 10% SDS-PAGE and transferred to PVDF membrane (Merck Millipore, Darmstadt, Germany). The membranes were blocked in 5% skimmed milk in Tris-buffered saline buffer with 1% Tween-20 for 1 h at room temperature, followed by incubation with primary antibodies overnight at 4 °C. Subsequently, the membranes were incubated with HRP-conjugated secondary antibodies, detected by enhanced chemiluminescence (ECL) solution (Pierce, Thermo Fisher Scientific, Waltham, MA, USA) using a chemiluminescence detection system (Biorad, Berkeley, CA, USA). Antibodies used in the present study for Western blotting were as follows: anti-CD9 (sc-59140; Santa Cruz Biotechnology, Dallas, TX, USA), anti-CD63 (CBL553; Millipore, Darmstadt, Germany), anti-β-actin (Sigma Aldrich Corporation, St. Louis, MO, USA), anti-PKM (3186, Cell signaling, Carlsbad, CA, USA), anti-CTSB (31718, Cell signaling, Carlsbad, CA, USA), and anti-CTSD (69854, Cell signaling, Carlsbad, CA, USA).

### 2.4. Proteomics and Bioinformatic Analyses

We performed label-free proteomics analysis on AML EVs isolated from three different cell strains (OCI-AML-2, OCI-AML-3, HL-60). First, EVs from two independent biological samples from each cell line were lysed through sonication (Sonicator U200S control, IKA Labortechnik, Staufen, Germany) at 70% amplitude in a lysis buffer (urea 6 M in 100 mM Tris/HCl, pH = 7.5). Then, protein concentration was measured by Bradford assay (Bio-Rad, Hercules, CA, USA) using Bovine Serum Albumin (BSA, Sigma-Aldrich, St. Louis, MI, USA) as standard for the calibration curve in order to digest 25 μg of proteins for each EV sample, obtaining three different pools (EVs OCI-AML-2, EVs OCI-AML-3 and EVs HL-60), as reported in our latest work [22]. Single-pot solid-phase-enhanced sample preparation (SP3) protocol was conducted following Moggridge et al. [23] by combing two types of carboxylate-functionalized beads (1:1) (Sera-Mag Speed Beads, GE Life Sciences, Cytiva, Marlborough, MA, USA, Part No: 45152105050350 and 65152105050350), which were washed twice with water, using a magnetic rack, before adding the lysed EVs at a ratio of 1:10 (μg protein/μg SP3 beads). An overnight tryptic digestion at 37 °C was carried out to obtain tryptic peptides that were acquired in triplicate by nanoLC-MS/MS using the UltiMate^TM^ 3000 UPLC (Thermo Fisher Scientific, Milan, Italy) chromatographic system coupled to the Orbitrap Fusion^TM^ Tribrid^TM^ (Thermo Fisher Scientific, Milan, Italy) mass spectrometer, as recently published [22]. MS/MS raw files were processed by MaxQuant, version 1.6.6.0, (Max-Planck Institute for Biochemistry, Martinsried, Germany) against the UniProt database (released 2020_06, taxonomy Homo Sapiens, 20,588 entries) to obtain iBAQ values used for bioinformatics, and functional analyses were performed using Perseus software, version 1.6.10.50, (Max-Planck Institute for Biochemistry, Martinsried, Germany) and Ingenuity Pathway Analysis software version 6.3 (IPA, Qiagen, Hilden, Germany). Details of all processing parameters have been described in previous works [24,25,26]. In particular, at protein level comparison we performed an ANOVA test with Benjamini–Hochberg FDR correction by Perseus version 1.6.10.50 to emphasize the significative EV proteins between the three AML-derived EVs. STRING (https://string-db.org/, accessed on 7 June 2025) analysis was used for evaluation of Protein–Protein Interaction (PPI) networks. FunRich (Functional Enrichment analysis tool, http://www.funrich.org/, accessed on 7 June 2025) was used to easily match our Leuko EV datasets with Vesiclepedia [27]. The PANTHER Classification System 18.0 was used for Gene Ontology protein reclassifications of EV proteins unmatched with Vesiclepedia.

## 3. Results

### 3.1. Extracellular Vesicles Isolation, Validation, and Characterization

We aimed to identify protein pattern profiles in EVs that may be used to characterize human AML cells. OCI-AML-2, OCI-AML-3, and HL-60 cell lines, commonly used to study AML, were selected to analyze vesicular protein outfit using a label-free proteomics approach. We isolated EVs from cellular supernatants by the ultracentrifugation method, as described in material and methods. To demonstrate the EVs enrichment in our fractions, we performed Western blot analysis for vesicular markers, according to the MISEV2018 and MISEV2023 guidelines [28,29]. As shown in Figure 1, Western blot analysis confirmed that vesicular lysates from OCI-AM-2 and OCI-AML-3 are highly enriched for CD9 and CD63, tetraspanin proteins, as for Alix, a well-known marker protein for EVs. As previously reported, unstimulated HL-60 cells did not express CD9 in whole lysate and EVs (Figure 1A,B) [30]. Indeed, it is well known from the literature that CD9 expression increased in HL-60 only after treatment with 12-O-tetradecanoylphorbol-1 3-acetate (TPA) [30]. Furthermore, as expected, the EVs preparations were negative for the Golgi marker GM130 (Figure 1B) and endoplasmic reticulum protein calnexin (Appendix A). In order to investigate the dimensions and size distribution of the isolated vesicles, DLS analysis was performed on diluted EVs suspensions. DLS revealed that EVs isolated from the three different cell lines exhibited mean diameters within the expected nanoscale range. EVs mean diameter varied among different cells, ranging from 300 nm to 360 nm. Notably, all three samples exhibited relatively high PDI values, (~0.3), suggesting sample heterogeneity. As also illustrated in Figure 1C, the intensity-based multimodal size distribution revealed the presence of two main populations: one centered around 100–200 nm and another between 600 and 1200 nm. This observation is consistent with the typical heterogeneity of EVs preparations obtained through ultracentrifugation-based extraction methods, which are known to co-isolate vesicles of different sizes without achieving strict size separation [31]. In our samples, despite the presence of the higher-size fraction, the subpopulation of smaller vesicles is clearly visible and, according to DLS theory, is known to represent the most abundant population in number. For further analysis, the surface zeta potential of all the extracted EVs was determined as a crucial parameter for the colloidal stability and aggregation tendency of EVs suspensions. Zeta potential measurements indicated that all EVs samples carried a negative surface charge, with values of −37.97 mV (OCI-AML-2), −29.17 mV (OCI-AML-3), and −32.12 mV (HL-60). The relatively high negative zeta potential in all samples suggests good colloidal stability of the EVs suspensions, since the EVs tend to remain dispersed in the solution rather than aggregating.

### 3.2. Proteome Profiling of AML-Derived EVs

To further characterize EVs 25 μg of each vesicular fraction was subjected to proteomics characterization. A total of 1129 proteins for EVs from HL-60 cells, 1352 proteins for EVs from OCI-AML-2 cells, and finally, 1370 proteins for EVs from OCI-AML-3 cells were quantified using proteomics analysis. Of these, 1004 proteins (67.5%) were in common between the three investigated samples, while 49 (3.28%), 36 (2.41%), and 57 (3.82%) were uniquely identified in EVs HL-60, EVs OCI-AML-2, and EVs OCI-AML-3, respectively (Figure 2A). The whole list of quantified proteins is reported in Appendix A. As reported in Appendix A, common proteins were interconnected in a single functional network (PPI enrichment *p*-value < 1 × 10^−16^), where Cellular Component STRING analysis revealed that the majority of proteins (*n* = 535/1004) were significantly involved in “Extracellular Exosome” (GO: 0070062, FDR = 2.56 × 10^−223^), confirming EV protein origin and EVs isolation. Similarly, when EV proteins were compared to Vesiclepedia (http://www.microvesicles.org, accessed on 7 June 2025), a free compendium of EV molecular data [27], more than 95% of the proteins were matched in this repository for each cell strain, so may be referred to as the extracellular region (Figure 2B). All the identified proteins unmatched with Vesiclepedia database are listed in Appendix A. As shown in Figure 2C, Molecular Function Reclassification in the PANTHER GO database was able to mainly reclassify them into catalytic (GO: 0003824, 42.9%) and binding (GO: 0005488, 28.6%) activities, following the identification of proteins belonging to “metabolite interconversion enzyme” (PC00262) and “protein-modifying enzyme” (PC00260) protein classes. To analyze EVs’ protein cargo across AML cells, we performed an expression analysis using IPA software (v. 6.3) with quantitative data obtained by MaxQuant. An overview of Canonical Pathways for EVs HL-60 is shown in Figure 2D as bubble charts. The most significantly modulated pathways are: “Neutrophil Degranulation” (−Log(*p*-value) = 83.5), “Eukaryotic Translation Initiation” (−Log(*p*-value) = 98.9), and “EIF2 Signaling” (−Log(*p*-value) = 88.7) for all three lysed EVs, as shown in the Appendix A for OCI-AML-2 and OCI-AML-3 EVs too (Appendix A). In particular, Neutrophil Degranulation is enclosed in Immune System Categories, suggesting that EVs are key players in the immune response exerted by leukemic cells within the tumor microenvironment. The role of EV proteins in Eukaryotic Translation Initiation, enclosed in ‘Metabolism’ of the proteins categories, highlighted the great ability of EVs to target specific reprogramming information as well as EIF2 Signaling, enclosed in the Signaling Pathway Category, which are involved in cellular growth, proliferation and development, cellular stress, and injury. The complete list of Canonical Pathways is reported in the Appendix A.

### 3.3. Evaluation of Selected Protein Biomarkers for AML-Derived EVs

The content of EVs is crucial to address a lot of functional information, so, according to the proteomics results, we thoroughly investigated the quantitative expression levels in some selected putative AML biomarkers. Among proteins identified as commonly expressed in EVs from all three AML cell lines, pyruvate kinase 1/2 (PMK1/2) was significantly higher in OCI-AML-2 and OCI-AML-3 than in HL-60, as shown in Figure 3A, where the relative abundance of the protein was measured as an iBAQ average of the analytical triplicate for each EVs sample. These data were confirmed by Western blot analysis, which reported the expression of the enzyme in EVs and in whole lysates, as shown in Figure 3B,C. Furthermore, we investigated cysteine cathepsin B (CTSB) and aspartic cathepsin D (CTSD) for their novel biological functions in secretory vesicles. As shown in the mechanistic network of Figure 3D, CTSB, together with another 84 proteins, is mainly related to the modulation of “binding of tumor cell lines” (*p*-value = 1.06 × 10^−17^), demonstrating that protein cargoes of AML-derived EVs are fundamental for tumor progression, since increased levels of CTSB circled in the network correlate with the metastatic process. To gain further insight we also performed Upstream Regulator Analysis within IPA, to highlight the probable causes that have induced observed expression changes. CTSD and CTSB were predicted as significant Upstream Regulators for all three investigated AML-EVs by the same protein dataset, as described in the functional network of Figure 3E (CTSB: EVs HL-60 *p*-value = 9.84 × 10^−3^, EVs OCI-AML-2 *p*-value = 4.54 × 10^−3^, EVs OCI-AML-3 *p*-value = 4.84 × 10^−3^; CTSD: EVs HL-60 *p*-value = 1.66 × 10^−5^, EVs OCI-AML-2 *p*-value = 3.39 × 10^−5^, EVs OCI-AML-3 *p*-value = 3.58 × 10^−5^). In this context, in order to confirm CTSs identification and functional characteristics for each leukemia cell line, we performed Western blot analysis both on whole lysates and EVs (Figure 3B,C).

CTSs are synthesized as pre-pro-cathepsins in the endoplasmic reticulum, trafficked via the Trans-Golgi network, and are released in the extracellular compartment both in the immature and active form enclosed in EVs [32]. The maturation process for CTSs also involved modification in the glycosylation state corresponding to changes in molecular weight [32,33]. Furthermore, increased glycosylation levels for CTSs are reported in various cancers and represent a risk factor for the metastatic process [32,33,34,35,36]. Interestingly, our data shows that CTSD and CTSB are present both as mature and immature forms in the vesicular compartment, as indicated from bands at ~48 and ~30 KDa (Figure 3B,C) for OCI-AML-3 and HL-60 cell lines. Conversely, in the whole lysate, CTSB was expressed in a mature form in all leukemia cells, while CTSD was expressed mainly as mature, with few amounts present in an immature form in each of the cell lines. Interestingly, OCI-AML-2 CTSB was expressed in whole cell lysates (Figure 3B) but was not released in EVs (Figure 3C). The complete list of downstream effects and Upstream Regulators with their relative overlap *p*-value is reported in Appendix A.

## 4. Discussion

Great interest in the EVs field is related to the protein profile of cancer cell-derived EVs as a promising strategy for the development of novel diagnostic and therapeutic approaches [37]. Furthermore, EVs possess biological properties of their parental cells and thus they have been successfully applied as therapeutic targets and agents [38]. Proteomic studies are useful tools to evaluate and identify biomarkers which can be used as potential signatures for non-invasive liquid biopsies and/or stratification between different cancer subtypes. A proteomic approach has already been used by Kang and colleagues to characterize AML cell-derived EVs from HL-60, KG-1, and THP-1 cell lines that mainly focus their attention on CD53 and CD47 as useful biomarker for AML disease [39].

In order to identify the proteomic profile of EVs derived from AML cells, we proceeded to isolate EVs from three human AML cells, OCI-AML-2, OCI-AML-3, and HL-60EVs, each characterized by a distinctive genotype. As suggested by the ISEV guideline, [28,29] we evaluated the exosome expression markers CD9, CD63, and Alix in EVs and in the whole cell lysate by Western blot (Figure 1A,B). CD63 showed remarkable differences in molecular weight between AML cell lines and EVs derived (Figure 1A,B). The tetraspanin CD63 is a highly N-glycosylated protein, and its surface expression is regulated by its glycosylation state [40,41]. Indeed, several studies reported a different CD63 expression pattern in Western blot analysis between cell lysate and EVs, suggesting that the smeared band represents the variable glycosylated forms of CD63 protein [42,43]. Interestingly, our data demonstrated that the N-glycosylation state of CD63 differs not only between cells and EVs, but also among the cell lines studied (Figure 1A,B). Using a gel-free MS-based proteomics approach, we analyzed EVs and detected a total of 1004 EV-associated proteins commonly expressed in all leukemia cell lines and 49, 36, and 57 EV-associated proteins quantified as unique in HL-60, OCI-AML-2, and OCI-AML-3, respectively (Figure 2A). As a proof-of-concept, we demonstrated that common proteins were mainly associated with the “Extracellular Exosome” (GO:0070062) (Appendix A) cellular component, and by matching our protein dataset with the Vesiclepedia database we also highlighted that more than 95% of the quantified proteins for each cell line were already associated with EV cargo (Figure 2B). In particular, EVs purified from all three AML cell lines were significantly enriched with proteins that control immune response, proliferation process, and metabolism (Figure 2D). The key role of PKM in energy processing as a key rate-limiting enzyme in glucose metabolism is well known [44]. Interestingly, our proteomic data and Western blot analysis show PKM protein expression in the EV compartment of all cell lines investigated (Figure 3A–C). PKM-EVs may interact with membrane lipids in recipient cells and thus control the metabolic processes in non-canonical manners [34,44]. Moreover, the vesicular PKM may enhance not only glycolysis but also drug resistance, as reported in several cancer types [45,46]. Furthermore, we focused on cathepsins (CTSs) expressed in the EV-associated leukemia cells for their novel biological functions in secretory vesicles [15,16]. In recent years, several studies have investigated the role of CTSs in tumorigenesis and their potential utility as pharmacological targets [17,18]. CTSB is the best-characterized member of the cysteine cathepsin family and is ubiquitously expressed in most cells [47,48]. Previous studies reported that CTSB increased levels correlate with cancer progression and metastatic processes [18,19,20]. CTSB has different subcellular localizations, it is mainly located in secretory vesicles and is secreted into the pericellular compartment [18,20,21]. Several studies have shown increased levels of circulant CTSB in plasma, sera, and cerebral spinal fluid related to neurodegenerative disorders and several solid tumors [16,22,23,24,25,26]. Furthermore, in pediatric AML patients enhanced enzymatic activity of CTSB has been reported and correlated with poor event-free survival (EFS) [27]. CTSD is an aspartic protease mainly involved in degrading unfolded proteins and that controls several processes in cancer progression, i.e., angiogenesis, invasion, metastasis, and drug resistance [28,29,30]. CTSD is mainly expressed in lysosomes, but CTSD and pro-CTSD are also released in the extracellular compartments from cancer cells to stimulate tumoral and stromal cells in the tumoral microenvironment [31]. In physiological conditions, pro-CTSD is found in the intracellular space, while it has been shown that, in pathological conditions, the pro-CTSD/CTSD levels ratio correlates with tumor aggressiveness and prognosis [31,32]. Cancer cells can release an amount of pro-CTSD in the extracellular space which can act as an autocrine and paracrine factor to promote cell survival [33]. Moreover, the ratio between pro-CTSD and CTSD has been related to metastasis-free survival and disease-free survival in breast cancer patients [31,34]. Our data showed a characteristic pro-CTSB expressed exclusively in the EVs released from OCI-AML-3 and HL-60 and also an increased pro-CTSD/CTSD in the EV compartment for all cell lines investigated (Figure 3B).

The distinctive patter of immature form of CTSB and D in EVs compartment could reflect the different genetic mutations in the three AML cell lines investigated and also correlate with the subtypes in the French-American-British (FAB) classification. Indeed, OCI-AML-2 and OCI-AML-3 cells were isolated from patients diagnosed with FAB subtype M4 characterized by an increased number of immature granulocytic and monocytic cells while HL-60 cells from a patient diagnosed with M3 characterized by an increased number of promyelocytic cells [49]. Furthermore, immature forms of CTSs could be released from AML cells, transported via EVs and then activated in the tumor environment, thus modulating pro-tumor mechanisms such as drug resistance and inflammation [50]. We might speculate that an additional activity of CTSs inhibitors in AML could also be linked to disrupting EVs formation and/or composition thus limiting AML-derived EVs activities in the BM niche. However, further studies have to be carried out to better understand secretory pathways for CTSs release from tumor cells and EVs-mediated mechanisms in AML disease.

## Figures and Tables

**Figure 1 cells-14-01651-f001:**
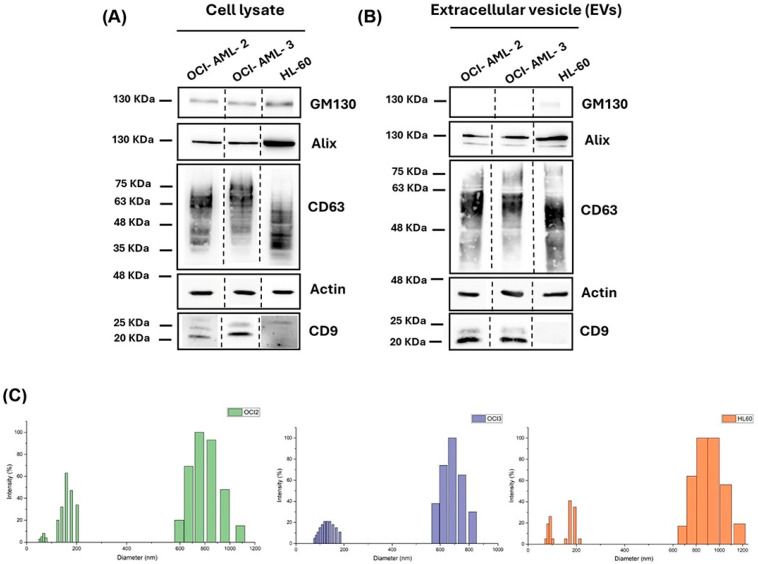
Characterization of EVs isolated from acute myeloid leukemia (AML) cells. (**A**,**B**) Western blot analysis of EVs markers CD9, CD63, Alix, and actin in leukemia whole cell lysates (**A**) and in EV lysates (**B**). (**C**) Size distribution of EVs derived from OCI-AML-2 (green graph), OCI-AML-3 (violet graph), and HL-60 and (orange graph).

**Figure 2 cells-14-01651-f002:**
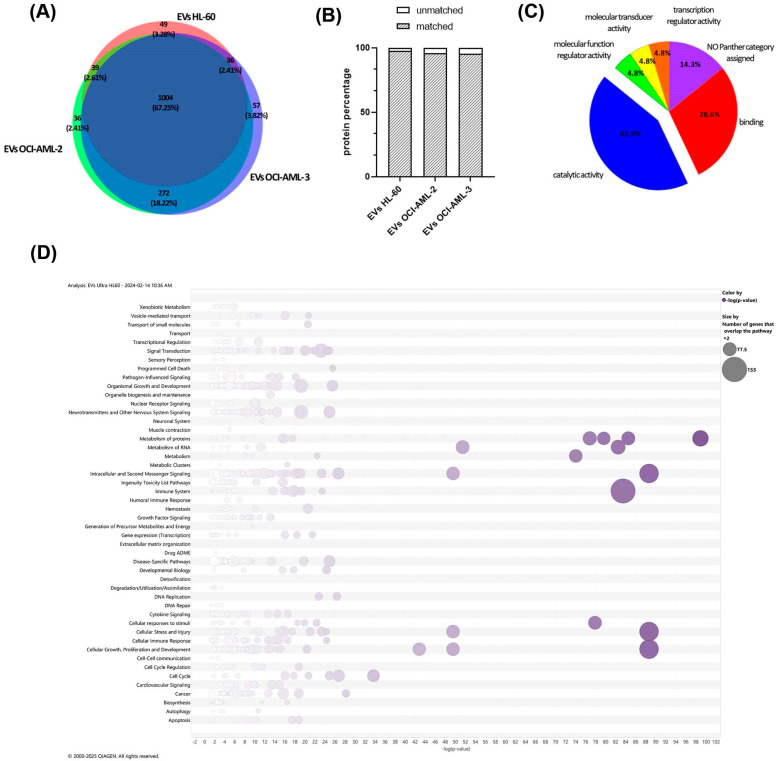
Proteomic characterization of EVs. (**A**) Venn Diagram of quantified proteins in the three AML EVs. (**B**) Proteomics analysis reveals a strong overlap of our dataset with Vesiclepedia, an online EV proteome repository. Only 2.54%, 4.29%, and 4.75% of the EV proteins were unmatched in Vesiclepedia for HL-60, OCI-AML-2, and OCI-AML-3 EVs, respectively. (**C**) PANTHER Molecular Function Reclassification analysis of EV proteins unmatched with the Vesiclepedia repository database. (**D**) Bubble charts represent Canonical Pathways significantly modulated by EVs-HL60 proteins. Bubble sizes are proportional to the number of proteins that overlap that pathway. B-H Multiple Testing Correction *p*-value: display only entities that have a −Log(*p*-value) greater than 1.3. *Y*-axis: Pathway Categories that include each single pathway. *X*-axis: −Log(*p*-value).

**Figure 3 cells-14-01651-f003:**
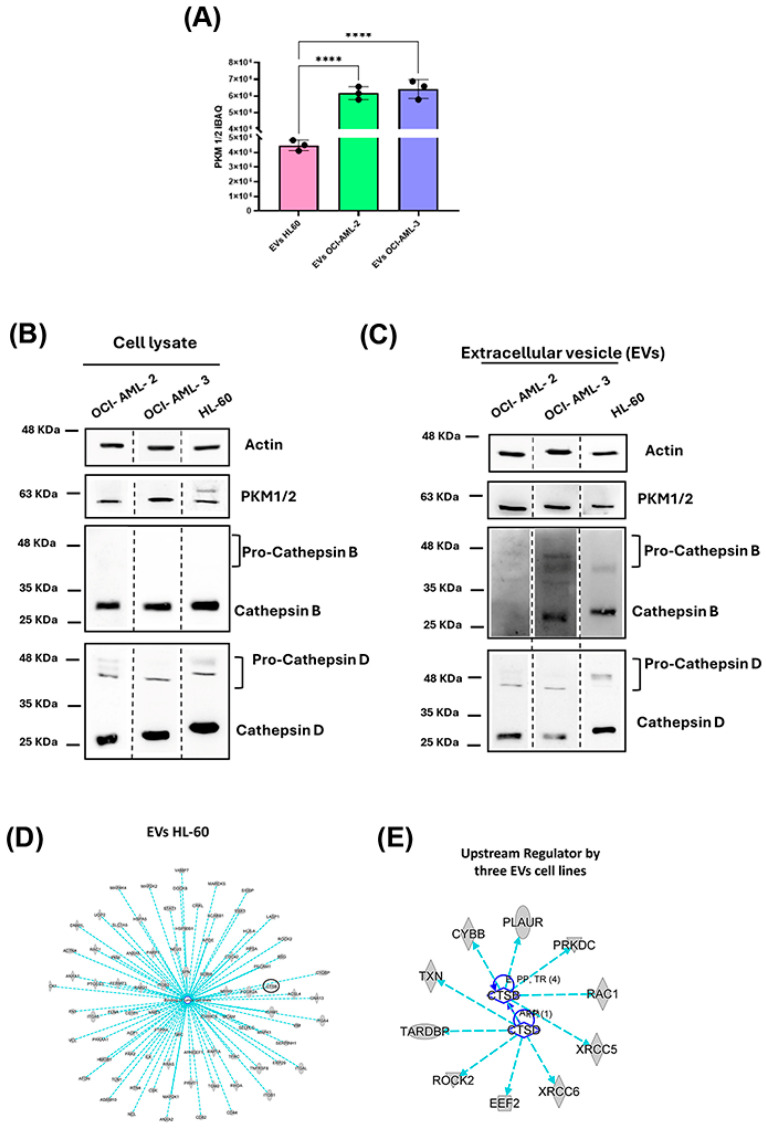
Validation of specific proteins in EVs isolated from acute myeloid leukemia (AML) cells. (**A**) Histogram representing the proteomics quantitative data related to PKM ½ abundance in EV samples highlighted by iBAQ value. ****: *p*-value < 0.0001 at ANOVA test and validated with post hoc Tukey’s HSD test. Data were reported as mean ± standard deviation of iBAQ of analytical replicates (*n* = 3). (**B**,**C**) Western blot analysis confirms proteomic data of PKM1/2, CTSB, and CTSD in leukemia whole cell lysates (**B**) and in EV lysates (**C**). Specific signals were detected for pro-cathepsin B and pro-cathepsin D (47 kDa band) and mature single-chain cathepsin B and D (30 kDa band). (**D**) Functional network of “binding of tumor cell lines” (*p*-value = 1.06 × 10^−17^), a significative function modulated by 85 EVs HL-60 proteins quantified by MS/MS. (**E**) Mechanistic networks of Cathepsin B (*CTSB*) and Cathepsin D (*CTSD*), two Upstream Regulators significantly modulated by EV proteins extracted from HL-60, OCI-AML-2 and OCI-AML-3.

## Data Availability

The original contributions presented in this study are included in the article/Appendix A. Further inquiries can be directed to the corresponding author.

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
