# Peer review of "Extracellular Vesicles Profiling in Acute Myeloid Leukemia Cell Lines: A Proteomic Characterization"

_cells, 2025, doi:10.3390/cells14211651_

Round 1

Reviewer 1 Report

Comments and Suggestions for Authors

The authors conducted a study aimed at exploring the proteome of extracellular vesicles (EVs) secreted by three different acute myeloid leukemia (AML) cell lines that differ in their genetic backgrounds. They collected EVs using a classic ultracentrifugation protocol and profiled them through an LC-MS/MS approach. A similar study was performed by Kang et al. (Molecular & Cellular Proteomics, 2021) using slightly different cell lines. In the present study, the authors identified pyruvate kinases 1/2 and cathepsins, which were differentially expressed between the selected cell lines. Although the overall novelty of the study is limited, it may still provide valuable information for researchers interested in the role of EVs in AML. Suggestions are listed below.

  1. Although the authors mentioned the MISEV guidelines, they did not fully adhere to them. Specifically, they did not include three positive markers and one negative marker for Western blot analysis. Additionally, they did not characterize EVs using nanoparticle tracking analysis or electron microscopy. These characterizations are essential for any EV-related study.
  2. In Figures 1A and 1B, the size of CD63 bands varies. What is the explanation for this observation? The authors should address this. Similarly, in Figures 2B and 2C, the sizes of Cathepsin B and D bands differ between cell lines. What are the underlying reasons for these differences?
  3. In the article by Kang et al., the authors profiled EVs secreted by HL-60 cells, but the proteins they highlighted differ from those identified in this manuscript. The authors should discuss the potential reasons for these differences.
  4. One of the key questions raised in the introduction is whether the proteome of secreted EVs varies in cells with different genetic alterations. However, the authors did not link their findings to the genetic backgrounds of the cell lines. A more careful discussion of this aspect could significantly enhance the value of the article.
  5. In figure 1B, it should be E”V”s.
  6. In Figures 1C, 1E, 1F, and 1G, the font size is very small and difficult to read. The authors should increase the font size to improve readability.

Author Response

Dear Reviewers,

We thank you for your constructive criticisms and suggestions, which allowed us to significantly improve the manuscript. We hope the manuscript can be now accepted in this revised form.

Changes responding to all the Reviewers suggestions and concerns are in trace mode in the revised text.

Please find here our point-by-point answers:

Reviewer 1

Q1. Although the authors mentioned the MISEV guidelines, they did not fully adhere to them. Specifically, they did not include three positive markers and one negative marker for Western blot analysis. Additionally, they did not characterize EVs using nanoparticle tracking analysis or electron microscopy. These characterizations are essential for any EV-related study.

Answer 1. We agree with the points raised by the Reviewer, consequently we improved the EVs characterization by adding two negative markers (GM130 in Figure 1 and Calnexin in Supplementary Figure 1) and one additional positive marker (Alix in Figure 1) to the already analyzed positive markers (CD63 and CD9).

Furthermore, we used Dynamic Light Scattering (DLS) to investigate the particle size distribution, the concentration and the z-potential values of EVs. Consequently, we added paragraph 2.2 on DLS in Materials and Methods and modified the Results section with DLS data, accordingly (Figure 1).

Q2. In Figures 1A and 1B, the size of CD63 bands varies. What is the explanation for this observation? The authors should address this. Similarly, in Figures 2B and 2C, the sizes of Cathepsin B and D bands differ between cell lines. What are the underlying reasons for these differences?

Answer 2. We thank the Reviewer for raising this important aspect of CD63 marker and cathepsins sizes, which are both linked with their glycosylation status.

CD63 is highly N-glycosylated protein and several studies have reported differences in the molecular weight of various glycosylated forms of CD63 (Tominaga N, Hagiwara K, Kosaka N, Honma K, Nakagama H, Ochiya T. RPN2-mediated glycosylation of tetraspanin CD63 regulates breast cancer cell malignancy. Mol Cancer. 2014 doi: 10.1186/1476-4598-13-134. PMID: 24884960; Dey S, Basu S, Ranjan A. Revisiting the Role of CD63 as Pro-Tumorigenic or Anti-Tumorigenic Tetraspanin in Cancers and its Theragnostic Implications. Adv Biol (Weinh). 2023 Jul;7(7):e2300078. doi: 10.1002/adbi.202300078. Epub 2023 May 4. PMID: 37142558).

Indeed, processing of CD63 with endoglycosidase H (endo H) and PNGase F enzymes, to remove the structure of N-glycan, demonstrated the existence of different forms of the CD63 protein with various degrees of glycosylation and corresponding to different sizes (Ageberg M, Lindmark A. Characterisation of the biosynthesis and processing of the neutrophil granule membrane protein CD63 in myeloid cells. Clin Lab Haematol. 2003 Oct;25(5):297-306. doi: 10.1046/j.1365-2257.2003.00541.x. PMID: 12974720.)

Cathepsin B and D are synthesized as inactive pre-proenzymes and then modified in their N-glycosidic linked oligosaccharide chain. These post-translational glycosylation modifications are essential for cathepsins maturation and signaling. Furthermore, previous studies by using  N- PNGase F and Endo H enzymes  to remove the structure of N-glycan on cathepsins have shown that glycosylation is responsible for their different band sizes in Western Blots (Yang L, Zeng Q, Deng Y, Qiu Y, Yao W, Liao Y. Glycosylated Cathepsin V Serves as a Prognostic Marker in Lung Cancer. Front Oncol. 2022 Apr 13;12:876245. doi: 10.3389/fonc.2022.876245. PMID: 35494076; PMCID: PMC9043764; Iacobuzio-Donahue CA, Shuja S, Cai J, Peng P, Murnane MJ. Elevations in cathepsin B protein content and enzyme activity occur independently of glycosylation during colorectal tumor progression. J Biol Chem. 1997 Nov 14;272(46):29190-9. doi: 10.1074/jbc.272.46.29190. PMID: 9360997.)

To underline this aspect, we added these sentences in the main text:

 ‘CD63 showed remarkable differences in molecular weight between AML cell lines and EVs derived (Figure 1A, B). The tetraspanin CD63 is a highly N-glycosylated protein, and its surface expression is regulated by its glycosylation state [31,32]. Indeed, several studies reported a different CD63 expression pattern in western blot analysis between cell lysate and EVs suggesting that the smeared band represents the variable glycosylated forms of CD63 protein [33,34]. Interestingly, our data demonstrated that the N-glycosylation state of CD63 differs not only between cells and EVs, but also among the cell lines studied (Figure 1A, B)’.

CTSs are synthesized as pre-pro-cathepsins in the endoplasmic reticulum, trafficked via Trans-Golgi network and are released in the extracellular compartment both in the immature and active form enclosed in EVs [35]. The maturation process for CTSs involved also modification in the glycosylation state corresponding to changes in molecular weight [35,36]. Furthermore, increased glycosylation levels for CTSs are reported in various cancers and represent a risk factor for metastatic process [35–39].

Q3. In the article by Kang et al., the authors profiled EVs secreted by HL-60 cells, but the proteins they highlighted differ from those identified in this manuscript. The authors should discuss the potential reasons for these differences.

Answer 3.  We thank the Reviewer for this comment. It is important to emphasize that our proteomics strategy differs substantially from that applied in the study of Kang et al. because our analysis is based on a label-free quantification (LFQ) proteomics approach, whereas in the article cited by the reviewer the authors the authors employed the tandem mass tag (TMT) labeling, a chemical label for sample multiplexing mass spectrometry-based quantification analysis  - in this case TMT tag on lysine, followed by a high-pH fractionation of tryptic peptides. These two proteomics approaches differ significantly in terms of identification and especially quantification of proteins. As a matter of fact, TMT labeling can increase the detection sensitivity of certain highly hydrophilic analytes, such as phosphopeptides; as a result, TMT-based workflows typically generate more enriched peptide pool, which supports a better in silico protein quantification. In contrast, the LFQ approach used in our study offers a distinct analytical perspective without the use of chemical labelling by analyzing directly the tryptic digestion. Moreover, the digestion protocol and the quantity of digested proteins were different too. Specifically, FASP protocol was used by Kang et al. to digest 30 µg of proteins, instead we digested 25 µg of proteins by SP3. These two different sample preparations differ in terms of their chemistry, cleanup efficiency and peptide recovery which can influence peptide yield and digestion efficiency. Despite that and the different MS/MS data processing software (Proteome Discoverer used by Kang and MaxQuant in our draft), we tried to overlap our protein dataset with Kang’s article highlighting that about 35% of proteins are quantified in common between the two protein matrices. Between them PKM and cathepsin B and D were discussed in our article.

According with the Reviewer's comment, we decided to implement the discussion section by mentioning the data of Kang and colleagues. We added this sentence to the main text:

Proteomic approach has already been used by Kang and colleagues to characterize AML cell–derived EVs from HL-60, KG-1, and THP-1 cell lines that mainly focus their attention on CD53 and CD47 as useful biomarker for AML disease.’

Q4. One of the key questions raised in the introduction is whether the proteome of secreted EVs varies in cells with different genetic alterations. However, the authors did not link their findings to the genetic backgrounds of the cell lines. A more careful discussion of this aspect could significantly enhance the value of the article.

Answer 4. We thank the Reviewer for the opportunity to better describe our data.  We decided to implement the discussion section, mentioning the possible link with the genetic differences between AML-cell lines investigated. We added this sentence to the main text:

‘The distinctive patter of immature forms of CTSB and D in EVs compartment could reflect the different genetic mutations in the three AML cell lines investigated and also correlate with the subtypes in the French-American-British (FAB) classification. Indeed, OCI-AML-2 and 3 cells were isolated from patients diagnosed with FAB subtype M4 characterized by an increased number of immature granulocytic and monocytic cells while HL-60 cells from a patient diagnosed with M3 characterized by an increased number of promyelocytic cells’.

Q5. In figure 1B, it should be E”V”s.

Answer 5. We modified the text in figure 1B

Q6. In Figures 1C, 1E, 1F, and 1G, the font size is very small and difficult to read. The authors should increase the font size to improve readability.

Answer 6. We modified the font size for all figures, we hope they are better now.

Reviewer 2 Report

Comments and Suggestions for Authors

Manuscript summary

The authors investigate extracellular vesicles in acute myeloid leukemia, employing label-free proteomics to characterize EVS isolated from three distinct AML cell lines: OCI-AML-2 (DNMT3A mutation), OCI-AML-3 (NPM1 mutation), and HL-60 (p53-null, c-MYC amplified). EVs were isolated using differential ultracentrifugation and analyzed via LC-MS/MS, with Western blot validation of selected proteins. The study aimed to identify EV protein signatures that could illuminate AML disease mechanisms.

Data unavailable to reviewer:  Supplementary Table S1 data is missing

Adherence to MISEV Guidelines

Authors state that the MISEV (2018/2023) guidelines were followed; however, this was not the case. At least not totally.

            Perform Western blot analysis for negative markers including calnexin, GM130, and cytochrome c to demonstrate absence of cellular organelle contamination.

            Conduct nanoparticle tracking analysis (NTA) or dynamic light scattering (DLS) to determine particle size distribution and concentration.

            Report particle-to-protein ratios as recommended quality control metrics for EV preparations.

Western Blotting

Authors do not indicate the amount of protein loaded in each lane of the gel to be transferred in the western blotting. In lanes 111 to 113, we can find, “and equal amounts of proteins were separated on 10% SDS-PAGE and transferred to PVDF membrane (Merck Millipore, Germany)”.

The quality of the CD63 WB image is significantly lower than that of the other antibodies used. The smeared pattern shown is not consistent with any sample of cell lysates or EVs. It has been discussed in research protocols and forums (tetraspanin glycosylation) and deserves a brief discussion about it and possible implications for the results interpretation.

In Figure 2C, Cathepsin B in OCI-AML-2 does not seem to be detected. However, in Figure 2D, Cathepsin 2B accounts for about 60% in OCI-AML-2 EVS. The authors do not mention this result and therefore do not discuss it. Authors need to address this question from both a technical perspective and regarding the functional extrapolations discussed in the discussion section.

Lane 146: “as recently publishe” (mistyping)

Proteomic Analysis

            Biological replicates are an essential step in label-free proteomics. Reviewing the methodological section, it is unclear or poorly described how biological replicates (at least three) are used. It is only in the results section that it can be inferred that biological replicates were employed. Authors must clearly state in the methodological section the use of biological replicates.

            Lanes 252 – 253: “CTSB was quantified by MS/MS analysis as unique protein only in HL-60 EVs (Figure 2 F).” This phrase does not make sense to me, and it appears to refer to an incorrect figure. Figures 2F and 2G, even when zoomed in, do not allow for correlation with the information presented in the text. They need to be presented in higher quality.

            Regarding figure quality, the same applies to figures 1E and 1F.   

            The statistical analysis framework requires a thorough revision to handle the large multiple testing burden when comparing over 1,000 proteins across three conditions simultaneously. Although the study reports p-values with high precision (e.g., "p-value = 1.06 × 10^-17"), there is not enough evidence that proper family-wise error rate or false discovery rate corrections were applied to the protein-level comparisons (Benjamini & Hochberg, 1995). The pathway analysis mentions "B-H Multiple Testing Correction" with a -log(p-value) threshold of more than 1.3; however, it appears to only apply to pathway enrichment, not individual protein comparisons. Without solid multiple comparison correction at the protein level, the family-wise error rate is nearly certain, making most of the reported "significant" differences probably false positives. (Benjamini, Y., & Hochberg, Y. (1995). Controlling the false discovery rate: a practical and powerful approach to multiple testing. Journal of the Royal Statistical Society, 57, 289-300. https://doi.org/10.1111/J.2517-6161.1995.TB02031.X)

Summary and Recommendations

The manuscript offers an interesting label-free proteomics analysis of extracellular vesicles (EVs) from AML cell lines, aiming to identify disease-related protein signatures. While the study tackles a relevant question and employs appropriate methods, several critical issues need to be addressed to enhance its rigor and alignment with current standards.

  1. MISEV Guideline Adherence
    • Although the authors state compliance with MISEV2018/2023, key elements are missing:
    • Absence of negative controls for cellular contamination (e.g., calnexin, GM130, cytochrome c).
    • No particle size/concentration assessment (e.g., NTA or DLS).
    • Lack of particle-to-protein ratio reporting.

  1. Western Blotting Quality and Interpretation
    • Protein loading amounts must be clearly stated in all figures and legends.
    • CD63 bands appear smeared and inconsistent; a discussion of tetraspanin glycosylation and its implications is warranted.
    • Cathepsin B inconsistencies (e.g., absent in OCI-AML-2 lysates, abundant in EVs) are not acknowledged and require explanation.
  2. Proteomic Analysis Transparency
    • The use of biological replicates should be clearly described in the Methods section, not left for inference.
    • Figures 1E, 1F, 2F, and 2G are of insufficient resolution to verify the claims made; higher-quality images are required.
    • The statistical framework lacks detail and likely fails to address multiple testing correction at the protein level. Without this, the reported significance is questionable. Proper FDR or family-wise correction (e.g., Benjamini-Hochberg) must be implemented and explicitly reported.
  3. Minor Issues
    • Typo in line 146: “publishe” should be corrected.
    • Supplementary Table S1 was not available for review.

In summary, the manuscript requires substantial revision to meet EV research standards and ensure the robustness of proteomic conclusions. Clarifying the methodology, strengthening quality controls, and addressing statistical concerns are essential before proceeding to further consideration.

Author Response

Dear Reviewers,

We thank you for your constructive criticisms and suggestions, which allowed us to significantly improve the manuscript. We hope the manuscript can be now accepted in this revised form.

Changes responding to all the Reviewers suggestions and concerns are in trace mode in the revised text.

Please find here our point-by-point answers:

Reviewer 2

The authors investigate extracellular vesicles in acute myeloid leukemia, employing label-free proteomics to characterize EVS isolated from three distinct AML cell lines: OCI-AML-2 (DNMT3A mutation), OCI-AML-3 (NPM1 mutation), and HL-60 (p53-null, c-MYC amplified). EVs were isolated using differential ultracentrifugation and analyzed via LC-MS/MS, with Western blot validation of selected proteins. The study aimed to identify EV protein signatures that could illuminate AML disease mechanisms.

Data unavailable to reviewer: Supplementary Table S1 data is missing.

We upload again the Supplementary data, hopefully they will be now visible in the system.

Q1. Adherence to MISEV Guidelines. Authors state that the MISEV (2018/2023) guidelines were followed; however, this was not the case. At least not totally. Perform Western blot analysis for negative markers including calnexin, GM130, and cytochrome c to demonstrate absence of cellular organelle contamination.

Answer 1. We agree with the points raised by the Reviewer, consequently we improved the EVs characterization by adding two negative markers (GM130 in Figure 1 and Calnexin in Supplementary Figure 1) and one more positive marker (Alix in Figure1) to the already analyzed positive markers (CD63 and CD9).

Q2. Conduct nanoparticle tracking analysis (NTA) or dynamic light scattering (DLS) to determine particle size distribution and concentration.

Answer 2. According to the Reviewer suggestion, we used Dynamic Light Scattering (DLS) to investigate the particle size distribution, the concentration and the z-potential values of EVs. Consequently, we added paragraph 2.2 on DLS in Materials and Methods and improved the Results with DLS data (Figure 1).

Q3. Report particle-to-protein ratios as recommended quality control metrics for EV preparations.

Answer 3. We agree with the points raised by the Reviewer, we added information about count rate and protein concentration in Supplementary Material in Table 2.

Q4. Western Blotting. Authors do not indicate the amount of protein loaded in each lane of the gel to be transferred in the western blotting. In lanes 111 to 113, we can find, “and equal amounts of proteins were separated on 10% SDS-PAGE and transferred to PVDF membrane (Merck Millipore, Germany)”.

Answer 4. We thank the Reviewer, we modified the material and methods section, and we specified the total protein amount (15 µg) used for western blot analysis.

Q5. The quality of the CD63 WB image is significantly lower than that of the other antibodies used. The smeared pattern shown is not consistent with any sample of cell lysates or EVs. It has been discussed in research protocols and forums (tetraspanin glycosylation) and deserves a brief discussion about it and possible implications for the results interpretation.

Answer 5. We thank the Reviewer for raising this important aspect of CD63 marker and cathepsins sizes, which are both linked with their glycosylation status.

CD63 is highly N-glycosylated protein and several studies have reported differences in the molecular weight of various glycosylated forms of CD63 (Tominaga N, Hagiwara K, Kosaka N, Honma K, Nakagama H, Ochiya T. RPN2-mediated glycosylation of tetraspanin CD63 regulates breast cancer cell malignancy. Mol Cancer. 2014 doi: 10.1186/1476-4598-13-134. PMID: 24884960; Dey S, Basu S, Ranjan A. Revisiting the Role of CD63 as Pro-Tumorigenic or Anti-Tumorigenic Tetraspanin in Cancers and its Theragnostic Implications. Adv Biol (Weinh). 2023 Jul;7(7):e2300078. doi: 10.1002/adbi.202300078. Epub 2023 May 4. PMID: 37142558).

Indeed, processing of CD63 with endoglycosidase H (endo H) and PNGase F enzymes, to remove the structure of N-glycan, demonstrated the existence of different forms of the CD63 protein with various degrees of glycosylation and corresponding to different sizes (Ageberg M, Lindmark A. Characterisation of the biosynthesis and processing of the neutrophil granule membrane protein CD63 in myeloid cells. Clin Lab Haematol. 2003 Oct;25(5):297-306. doi: 10.1046/j.1365-2257.2003.00541.x. PMID: 12974720.)

To underline this aspect, we added these sentences in the main text:

CD63 showed remarkable differences in molecular weight between AML cell lines and EVs derived (Figure 1A, B). The tetraspanin CD63 is a highly N-glycosylated protein, and its surface expression is regulated by its glycosylation state [31,32]. Indeed, several studies reported a different CD63 expression pattern in western blot analysis between cell lysate and EVs suggesting that the smeared band represents the variable glycosylated forms of CD63 protein [33,34]. Interestingly, our data demonstrated that the N-glycosylation state of CD63 differs not only between cells and EVs, but also among the cell lines studied (Figure 1A, B)’.

Q6. In Figure 2C, Cathepsin B in OCI-AML-2 does not seem to be detected. However, in Figure 2D, Cathepsin 2B accounts for about 60% in OCI-AML-2 EVS. The authors do not mention this result and therefore do not discuss it. Authors need to address this question from both a technical perspective and regarding the functional extrapolations discussed in the discussion section.

Answer 6. We thank the Referee for raising this question. We decided to remove the graph 2D and E which could be misinterpreted, and we better described the data on CTSs expression and their possible role in AML disease. To underline this aspect, we added these sentences in the main text:

‘The distinctive patter of CTSB and D in EVs compartment could be correlated to the difference genetic mutations in the three AML cell lines investigated and also to the subtypes in the French-American-British (FAB) classification. Indeed, OCI-AML-2 and 3 cells were isolated from patients diagnosed with FAB subtype M4 characterized by an increased number of immature granulocytic and monocytic cells while HL-60 cells from patient diagnosed with M3 characterized by an increased number of promyelocytic cells [46]. However further studies are required to better to better understand and clarify this aspect.

 Q7. Lane 146: “as recently publishe” (mistyping)

Answer 7. We corrected the error

Q8. Biological replicates are an essential step in label-free proteomics. Reviewing the methodological section, it is unclear or poorly described how biological replicates (at least three) are used. It is only in the results section that it can be inferred that biological replicates were employed. Authors must clearly state in the methodological section the use of biological replicates.

Answer 8. We thank the Reviewer for this comment, and we apologize for the lack of clarity. We have now revised the methodological section specifying that we have obtained two independent EV lysates for each sample and then, we have pooled them for proteomics purposes obtaining three different samples called EVs OCI-AML-2, EVs OCI-AML-3 and EVs HL-60 that were analyzed in triplicate by nanoLC-MS/MS.

  Q9. Lanes 252 – 253: “CTSB was quantified by MS/MS analysis as unique protein only in HL-60 EVs (Figure 2 F).” This phrase does not make sense to me, and it appears to refer to an incorrect figure. Figures 2F and 2G, even when zoomed in, do not allow for correlation with the information presented in the text. They need to be presented in higher quality.

Answer 9. We apologize for the incorrect and misleading phrase that we have now rephrased. CTSB was quantified in our protein dataset, specifically as unique protein in HL-60 EVs sample and not in OCI-AML-3 EVs and OCI-AML-2 EVs. Simultaneously, we performed upstream regulator analysis by Ingenuity Pathway Analysis in order to study the factors which may be causing the observed EV proteins expression. The results highlighted that both CTSB and CTSD were modulated by protein EV cargo of all three analyzed AML cell lines. Moreover, we uploaded the figure with higher quality.

 Q10. Regarding figure quality, the same applies to figures 1E and 1F.   

Answer 10. We uploaded figures with higher quality

Q11. The statistical analysis framework requires a thorough revision to handle the large multiple testing burden when comparing over 1,000 proteins across three conditions simultaneously. Although the study reports p-values with high precision (e.g., "p-value = 1.06 × 10^-17"), there is not enough evidence that proper family-wise error rate or false discovery rate corrections were applied to the protein-level comparisons (Benjamini & Hochberg, 1995). The pathway analysis mentions "B-H Multiple Testing Correction" with a -log(p-value) threshold of more than 1.3; however, it appears to only apply to pathway enrichment, not individual protein comparisons. Without solid multiple comparison correction at the protein level, the family-wise error rate is nearly certain, making most of the reported "significant" differences probably false positives. (Benjamini, Y., & Hochberg, Y. (1995). Controlling the false discovery rate: a practical and powerful approach to multiple testing. Journal of the Royal Statistical Society, 57, 289-300. https://doi.org/10.1111/J.2517-6161.1995.TB02031.X)

Answer 11.  We thank the Reviewer for his/her valuable comments. To underline this aspect, we added these sentences in the main text:

‘In particular, at protein level comparison, we performed ANOVA test with Benjamini-Hochberg FDR correction by Perseus version 1.6.10.50 to emphasize the significative EV proteins between the three AML-derived EVs’.

 In summary, the manuscript requires substantial revision to meet EV research standards and ensure the robustness of proteomic conclusions. Clarifying the methodology, strengthening quality controls, and addressing statistical concerns are essential before proceeding to further consideration.

We thank the Reviewer for his/her constructive criticisms and suggestions, which allowed us to significantly improve the manuscript. We hope it can now be evaluated for further consideration.

Reviewer 3 Report

Comments and Suggestions for Authors

The Figure 1G there it is not able to read what is written, because letters are too small, it should be corrected. In the part of introduction and materials and methods there is missing a description of the used cells in the study. Per example HL-60 cells, which type of cells are? Why they were using HL-60, not some other cell lines? Explain...Graph in the Figure 2 is not clearly visible, the letters are too small, and also for other figures. List of abbreviations should be supplemented. 

Author Response

Dear Reviewers,

We thank you for your constructive criticisms and suggestions, which allowed us to significantly improve the manuscript. We hope the manuscript can be now accepted in this revised form.

Changes responding to all the Reviewers suggestions and concerns are in trace mode in the revised text.

Please find here our point-by-point answers:

Reviewer 3

Q1. The Figure 1G there it is not able to read what is written, because letters are too small, it should be corrected.

Answer 1. We thank the Reviewer, we modified the figure.

Q2. In the part of introduction and materials and methods there is missing a description of the used cells in the study. Per example HL-60 cells, which type of cells are? Why they were using HL-60, not some other cell lines?

Answer 2. We thank the Reviewer for the opportunity to better describe the cell lines used in this work.  The introduction section describing cell lines characteristics (line 64-75) has been improved with others cell lines characteristics in results section.

We added these sentences in the main text:

‘The distinctive patter of immature form of CTSB and D in EVs compartment could be correlated to the difference genetic mutations in the three AML cell lines investigated and also to the subtypes in the French-American-British (FAB) classification. Indeed, OCI-AML-2 and 3 cells were isolated from patients diagnosed with FAB subtype M4 characterized by an increased number of immature granulocytic and monocytic cells while HL-60 cells from patient diagnosed with M3 characterized by an increased number of promyelocytic cells [48]. Furthermore, immature forms of CTSs could be released from AML cells, transported via EVs and then activated in the tumor environment, thus modulating pro-tumor mechanisms such as drug resistance and inflammation. [49] We might speculate that an additional activity of CTSs inhibitors in AML could also be linked to disrupting EVs formation and/or composition thus limiting AML-derived EVs activities in the BM niche.’

Q3. Explain...Graph in the Figure 2 is not clearly visible, the letters are too small, and also for other figures.

Answer 3. We have changed all the figures, we hope they are better now.

Q4. List of abbreviations should be supplemented. 

Answer 4. We added the list of abbreviations.

Round 2

Reviewer 1 Report

Comments and Suggestions for Authors

I appreciate that the authors have carefully addressed the comments from the previous version. One final comment regarding the size distribution of EVs: there appear to be particles ranging from 600 to 1000 nm. This raises the question of whether their EV samples contain a large proportion of soluble proteins or protein aggregates. The authors should discuss this point, especially since the manuscript heavily relies on proteomics results derived from these samples.

Author Response

Reviewer 1

Q1. “I appreciate that the authors have carefully addressed the comments from the previous version. One final comment regarding the size distribution of EVs: there appear to be particles ranging from 600 to 1000 nm. This raises the question of whether their EV samples contain a large proportion of soluble proteins or protein aggregates. The authors should discuss this point, especially since the manuscript heavily relies on proteomics results derived from these samples”.

Answer 1. We thank the Reviewer for this suggestion. As reported in the EVs isolation, validation and characterization section of the manuscript, DLS measurements performed on EVs isolated from the three different cell lines showed an average hydrodynamic diameter of approximately 300-350 nm, with a relatively high polydispersity index, suggesting sample heterogeneity. The intensity-based multimodal size distribution revealed the presence of two main subpopulations: one centered around 100-200 nm and another between 600 and 1200 nm. According to DLS theory, larger particles scatter light much more intensely than smaller ones, and therefore their signal can dominate the overall distribution. This effect may lead to an apparent overestimation of larger particles, even when they are numerically less abundant. In our samples, despite the presence of this higher-size fraction, the subpopulation of smaller vesicles is clearly visible and is likely to represent the most abundant population in number. This observation is consistent with the typical heterogeneity of EVs preparations obtained through ultracentrifugation-based extraction methods, which are known to co-isolate vesicles of different sizes without achieving strict size separation. For example, as reported in a previous study, urinary EVs isolated by ultracentrifugation exhibited a broad size distribution, with a predominant population around 100-200 nm together with larger micrometric vesicles with globular shape observed by AFM analysis, confirming that such size heterogeneity is typical of EVs isolates (Pilato, S. Mrakic-Sposta, V. Verratti, C. Santangelo, S. di Giacomo, S. Moffa, A. Fontana, T. Pietrangelo, F. Ciampini, S. Bonan, P. Pignatelli, C. Noce, P. di Profio, M. Ciulla, D. Bondi,* F. Cristiano, Urineprint of high-altitude: insights from analyses of urinary biomarkers and bio-physical-chemical features of extracellular vesiclesBiophys. Chem., 316, 107351:1-10 (2025). In addition, it is possible that a minor fraction of non-vesicular impurities or aggregates co-precipitated with the vesicles and contributed to the DLS signal. However, since DLS does not represent a qualitative technique, it is not possible to distinguish whether the larger scattering species are impurities, protein aggregates or large phospholipid vesicles. To underline this aspect, we added these sentences in the main text:

Notably, all three samples exhibited relatively high PDI values (~0.3), suggesting sample heterogeneity. As also illustrated in Figure 1C, the intensity-based multimodal size distribution revealed the presence of two main populations: one centered around 100-200 nm and another between 600 and 1200 nm. This observation is consistent with the typical heterogeneity of EVs preparations obtained through ultracentrifugation-based extraction methods, which are known to co-isolate vesicles of different sizes without achieving strict size separation.1 In our samples, despite the presence of the higher-size fraction, the subpopulation of smaller vesicles is clearly visible and, according to DLS theory, is known to represent the most abundant population in number.